# Housing starts and the associated wood products carbon storage by county by Shared Socioeconomic Pathway in the United States

**Jeffrey P. Prestemon**[1]\*, **Prakash Nepal**[2], **Kamalakanta Sahoo**[2,3]

**1** USDA Forest Service, Southern Research Station, Research Triangle Park, North Carolina, United States of America, **2** USDA Forest Service, Forest Products Laboratory, Madison, Wisconsin, United States of America, **3** Department of Biological Systems Engineering, University of Wisconsin, Madison, Wisconsin, United States of America

\* jeffrey.prestemon@usda.gov

**Data Availability Statement:** Mortgage delinquency rate data cannot be shared because of copyright. Data are available from the Mortgage Bankers Association, https://www.mba.org/

## Abstract

Harvested wood products found in the built environment are an important carbon sink, helping to mitigate climate change, and their trends in use are determined by economic and demographic factors, which vary spatially. Spatially detailed projections of construction and stored carbon are needed for industry and public decision making, including for appreciating trends in values at risk from catastrophic disturbances. We specify econometric models of single-family and multifamily housing starts by U.S. Census Region, design a method for their spatial downscaling to the county level, and project their quantities and carbon content according to the five Shared Socioeconomic Pathways (SSPs). Starts are projected to decline across all scenarios and potentially drop to below housing replacement levels under SSP3 by mid-century. Wood products carbon stored nationally in structures in use and in landfills is projected to grow across all scenarios but with significant spatial heterogeneity related to disparate trends in construction across counties, ranging from strong growth in the urban counties of the coastal South and West to stagnation in rural counties of the Great Plains and the northern Rockies. The estimated average annual carbon stored in wood products used in and discarded from US residential housing units between 2015–2070 ranged from 51 million t $CO_2$e in SSP3 to 85 million t $CO_2$e in SSP5, representing 47% to 78% of total carbon uptake relative to uptake by all wood products in the United States in 2019.

## Introduction

The Intergovernmental Panel on Climate Change (IPCC) projected the consequences of climate change and transmitted what it considered to be the urgency of reducing net emissions of greenhouse gases [1]. In 2020, the building operations and construction industry accounted for nearly 38% of the total global $CO_2$ emissions associated with energy use [2]. However, the residential housing construction sector also plays a sequestering role in net $CO_2$ emissions when those constructions are built mainly of wood. For instance, more than 90% of new single-family homes in the United States are constructed mainly of wood. Future inventories of

Mortgage interest rate data are available from Freddie-Mac, http://www.freddiemac.com/pmms/pmms30.html U.S. gross domestic product and its deflator are available from the U.S. Department of Commerce, https://www.bea.gov/data/gdp/gross-domestic-product Housing starts and permits data are available from the U.S. Census Bureau, https://www.census.gov/econ/currentdata/dbsearch?program=RESCONST&startYear=1959&endYear=2020&categories=STARTS&dataType=TOTAL&geoLevel=US¬Adjusted=1&submit=GET+DATA&releaseScheduleId= U.S. population data by state (aggregable to Census Region) are available from the Census Bureau at https://www2.census.gov/programs-surveys/popest/datasets/ U.S. population and income projections by county are available from the S1 Dataset indicated at https://journals.plos.org/plosone/article?id=10.1371/journal.pone.0219242 Other relevant data on the projections are available within the article, its supporting information, and from the Contact author, without limitations.

**Funding:** This research was funded in part by the USDA Forest Service, Forest Products Laboratory, and the U.S. Endowment for Forestry & Communities under USDA Forest Service Joint Venture Agreement to KS [18-JV-11111137-021]." Please also update the current FI to the following: "USDA Forest Service Joint Venture Agreement - 18-JV-11111137-021 - Dr. Kamalakanta Sahoo.

**Competing interests:** The authors have declared that no competing interests exist.

residential housing in the United States and carbon stored in wood products in those structures will be determined by the net of new units built and those destroyed. Wood used in the 141 million existing housing units [3] and other end uses and wood discarded in solid wood disposal sites (SWDS) in the United States stored an estimated 9.8 and 9.9 billion tons of carbon dioxide equivalent ($CO_2$e) in 2019 and 2020, respectively. The annual changes in the harvested wood products (HWP) carbon stock between these two years was 110 million tons of $CO_2$e, representing about 16% of net $CO_2$ uptake (flux) from the entire U.S. forest sector in 2019 [4]. Because wood products store carbon for many decades, and because wood can replace carbon-intensive materials such as steel and concrete in construction, the forest products sector can play an important role in mitigating net carbon emissions [5–7]. New housing units are demanded in part to replace the annual loss of approximately 0.4 million housing units [8] due to natural disasters, decay, movement of mobile homes, and market factors (e.g., torn down to make way for new development) [9] and to accommodate a growing and increasingly wealthy population. Analysts interested in understanding the long-run potential carbon storage contribution of the residential housing sector (e.g., [10]) may benefit from the development of statistical models with few assumptions beyond assumed rates of income growth.

Research has shown that reduced-form models of quarterly aggregate total, single-family, and multifamily new units started in the United States can be explained with high precision using only the rate of growth in U.S. gross domestic product (GDP), the mortgage delinquency rate, and seasonal indicators [11]. The cited study also indicated that inclusion of an additional variable which describes the aggregate rate of population growth could improve the fit of estimated models but that population's statistical role was uncertain. The research included projections of housing starts to 2070 under alternative rates of GDP growth and a model of mortgage delinquencies that also depended on GDP growth. Montgomery ([12–14]) reported that U.S. population growth is an important predictor of the number of households (housing inventory stock). Her research would imply that models of households that included per capita economic growth but not population growth may not predict the same demand for new housing units as models that included population as a driver. Models such as those employed by Prestemon et al. [11], in their housing projections by GDP growth rate, would therefore overestimate residential construction if population growth were to decline; Japan offers a case (e.g., [15, 16]) illustrating how positive economic growth and a zero growth to shrinking population combine to put downward pressure on housing demand. For the United States, population projections are available from the U.S. Census Bureau [17] to 2060. Population and GDP projections are core components of the Shared Socioeconomic Pathways scenarios (SSPs) that are adjuncts to the Intergovernmental Panel on Climate Change's climate projections [18, 19]. SSPs represent contrasting world visions as described by varying assumptions about demographic, economic, technological, environmental, and policy futures, creating varying degrees of challenges for climate change mitigation and adaptation in individual countries [18, 19].

Because land available for housing is related to income earning opportunities that vary across space and over time, it follows that rates of change in construction should vary across space and over time, according to how population and income vary (e.g., [20]). Private sector projections of households (e.g., [21]) are made at the county level, but such projections are not offered under alternative scenarios of the future. New research that offers such scenario-based projections will be useful for those seeking to understand how demand for wood products for construction could evolve into the future across regions in the United States (e.g., [22]), paths of possible expansion of housing into the wildland at the county scale (e.g., [23]), and in the identification of the locations where housing growth could interact with growing rates of climate-driven natural disturbances (e.g., [24, 25]) and rising seas (e.g., [26]).

Prestemon et al. [11] projected housing starts and associated softwood lumber consumption to 2070 under varying rates of income growth. The authors' projection model, based on quarterly data, evaluated but rejected as non-significant the effect of contemporaneous changes in total population on total housing starts in the United States. The authors also found, however, that a longer run (a 5-year) change in population could be statistically significant, opening the door to the possibility that projections of housing starts could be improved with the inclusion of longer run changes in population, not just aggregate U.S. income. A possible explanation of the non-significance of the contemporaneous change variable, however, is that the data on population are reported annually by the U.S. Census Bureau, and that even aggregate population estimates are subject to error. In other words, algebraically smoothed estimates of population could more accurately reflect rates of change at finer temporal scales (e.g., the quarter), enabling the identification of the effect of the population change variable on housing starts. Another limitation of the latter article was the aggregate (total U.S.) spatial unit being modeled. More spatially disaggregated modeling might uncover spatial differences in data generation processes for housing starts, allowing for more accurate overall assessments of the effects of population and income changes at finer spatial scales, such as U.S. Census Regions and counties.

This research has two primary objectives. First, we seek to demonstrate how population and income changes can be combined to project rates of new housing construction at fine and aggregate spatial scales in the United States under alternative socioeconomic scenarios of the future. Second, we seek to couple housing futures with projections of carbon storage in the housing sector in the United States. To make our projections of new housing, we specify models of housing starts at the Census Region level in the United States. These models, expanding from Prestemon et al. [11], are reduced-form specifications of single-family and multifamily housing starts by Census Region that include population changes as well as changes in aggregate U.S. income and a limited set of additional variables that control for mortgage credit market factors. Projections of single-family and multifamily starts are made at Census Region and U.S. aggregate level from 2016 to 2070 by SSP, applying some of the methods used by other authors and the income and population projections reported by Wear and Prestemon [20]. Furthermore, projections at the Census Region level are downscaled to the counties within that region. We describe in this article a method for making unbiased projections at the county level using the Census Region parameters, based on historical housing permits. The result of this effort is to show how rates of new construction of single-family and multifamily units would change over space and over time, consistent with the income and population projections at those spatial and temporal scales. New construction at those scales is further described by measures of carbon stored in the wood products that go into and remain used in housing units in the United States, including carbon stored in wood used to repair and remodel, and carbon stored in wood that is discarded and landfilled after demolition of housing units. These housing and carbon projections can be used by industry and policy makers to better understand where shipments will be destined and how the role of residential construction in storing carbon could evolve into the future.

This paper is organized as follows: First, we briefly describe our theoretical and empirical models of housing starts. We then outline how housing starts are downscaled to the county level. We next describe the uncertainties in our starts projections with Monte Carlo methods. We outline how starts projections at the county level are used to project harvested wood products carbon stored in residential wood structures in each county, including carbon stored while the units are in use and after their discards.

## Materials and methods

The number of new housing units constructed can be described as the equilibrium quantity emerging from the equating of the demand for and the supply of new housing. Aggregate demand for new housing, $Q_D^H$, is described as a function of housing price ($P^H$), credit factors ($C^H$), income ($Y$), and population ($U$):

$$Q_D^H = f(P^H, C^H, Y, U) \tag{1}$$

Aggregate supply for new housing, $Q_S^H$, is a function of housing price ($P^H$), construction input prices ($W^H$), and exogenous factors, including building regulations ($Z^H$):

$$Q_S^H = g(P^H, W^H, Z^H) \tag{2}$$

At equilibrium, $Q_D^H = Q_S^H = Q^H$, so the quantity of houses built is derived by equating (1) and (2) and solving for $Q^H$ and factoring out housing price:

$$f(P^H, C^H, Y, U) = Q_D^H = Q^H = Q_S^H = g(P^H, W^H, Z^H)$$
$$Q^H = h(C^H, Y, U, W^H, Z^H) \tag{3}$$

Credit factors can include the mortgage interest rate and the rate of mortgages in delinquency [11]. In [11], it was shown that the delinquency rate served to capture the effects of credit access (e.g., [27]), while the mortgage interest rate can additionally account for loan accessibility by potential home buyers. Prices of construction input factors could be indexed by lumber and panel prices, concrete prices, and energy prices. Exogenous construction supply factors could be indexed by regional or other fixed effects or by the inclusion of a time trend if such factors are judged to be trending in any way. Income can be described as disposable personal income or more simply with GDP. Income can also be included by dividing GDP by population (e.g., Montgomery [13, 14]). Prestemon et al. [11] found that wood prices were not significant determinants of the number of new houses started. In the identification of a parsimonious version of a quarterly time series version of Eq (3), they additionally found that a well-fitting model could be specified as a function of only GDP, the national mortgage delinquency rate, quarterly dummies, and, recognizing the autoregressive time series properties of (3), the lagged number of housing starts. One model presented in Prestemon et al. [11] reported that total U.S. housing starts had slightly better fit and lower likelihood of significant residual serial correlation when including the change in the mortgage interest rate.

In this study, we specify a reduced-form model of quarterly U.S. housing starts by type (single-family, multifamily) [28] that is similar to Prestemon et al. [11]. Different from that study, and in the interest of refining our understanding of how housing construction may differ across the United States, models of single-family and multifamily starts are estimated for each of four U.S. Census Bureau regions (Northeast, South, Midwest, West) [28] and include regional or national population as a predictor [17, 29], recognizing the differential roles of income and population change on housing demand [13]. We additionally specify (1) a reduced-form quarterly model of the percentage of mortgage delinquencies [30, 31], (2) a reduced-form quarterly model of the mortgage interest rate [32], and (3) a reduced-form model of national GDP [33] growth that captures the temporal dynamics of that variable. Census-reported historical population data were annual observations, which we smoothed for use in this study by converting them first to quarterly through interpolation and then by generating an equal-weighted centered moving average smoothing of the form,

$$U_t = \sum_{i=-u}^{u} U_{t+i} \Big/ 2u + 1), \text{ where } u = 9. \text{ The smoothing was applied to the total U.S.}$$

population and to Census Region population estimates for equations that contained the Region population as a regressor.

All dependent variables are projected jointly to 2070, with exogenous projections of variables in the reduced-form equations provided by Wear and Prestemon [20]. Multiple functional forms of housing starts models, including the same variables, are estimated and evaluated, including log-linear least squares and Poisson Pseudo Maximum Likelihood (PPML) models, which include various transformations of population.

Projections of housing starts by type by Census Region are done with Monte Carlo methods that match those of Prestemon et al. [11]. Briefly, models are estimated with a random sample with replacement of historical data on predictor variables from 1979 to 2016 and then projected to 2070 by quarter and summarized annually. GDP and population projections for Census Regions match U.S. regional or national GDP and population projections by SSP.

Downscaling of housing starts projections to the county level is done using the equation estimates for housing starts and the rates of change in county disposable personal income per capita or disposable personal income and population as reported by Wear and Prestemon [20]. Because the number of housing starts at the county level is unknown, starts at the county level are proxied by annual housing permits in the historical data [28]. Although county housing permit data are available on a monthly basis, permits precede starts, so that annual totals of permits in a county are likely to be more closely aligned with annual starts in the county. But because the starts models estimated in our current study are based on quarterly data, we created, for each county, historical pseudo-time series of quarterly housing permits for single-family and multifamily units. The annual permit observations, 2000–2015, were converted to quarterly permits by applying the following equation to all counties, 2000–2015:

$$h_{t,q,R,m}^{\tau} = [h_{t,R,m}^{s\tau}\exp(\hat{\beta}_{q,R}^{\tau}) \cdot \sum_{q=1}^{4}\exp(\hat{\beta}_{q,R}^{\tau})] + \delta \tag{4}$$

Where $h_{t,q,R,m}^{\tau}$ is the number of housing permits of type $\tau$ (single-family, multifamily) in year $t$ in quarter $q$ in Census Region $R$ and county $m$. Parameters $\hat{\beta}_{q,R}^{\tau}$ are the quarterly (seasonality) parameter estimates for housing of type $\tau$ in quarter $q$ in Census Region $R$. Finally, $\delta$ is set at 0.001, replacing zero for counties with minimal building activity (needed for the calculation of logarithmic errors of the housing starts estimate). Note that $\hat{\beta}_{4,R}^{\tau} = 0$ in Eq (4) (i.e., the fourth quarter indicator) was dropped in the Census Region starts models, which included an intercept term.

To predict quarterly permits (starts) at the county level in the historical time series, 2001–2015, the following equation was applied:

$$\ln\left(\hat{H}_{t,q,R,m}^{\tau}\right) = c_{R,m}^{\tau} + \mathbf{x}_{t,q,R,m}^{\tau\prime}\hat{\boldsymbol{\beta}}_{R}^{\tau} \tag{5}$$

Where ln is the natural logarithm operator, $\hat{H}_{t,q,R,m}^{\tau}$ is the predicted number of housing starts of type $\tau$ (single-family, multifamily) in year $t$ in quarter $q$ in Census Region $R$ and county $m$; $c_{R,m}^{\tau}$ is the constant (fixed effect) for the county, $\mathbf{x}_{t,q,R,m}^{\tau\prime}$ is a vector containing $l\,(\leq 4)$ lagged predicted starts ($\hat{H}_{t-l,q,R,m}^{\tau}$), quarterly seasonal indicators, the quarterly rate of change in the county's disposable income per capita (single-family starts models only) or income (multifamily starts models only), the quarterly change in population (single-family starts models only), the natural log of the national total mortgage delinquency rate, and the natural log of the mortgage interest rate. $\hat{\boldsymbol{\beta}}_{R}^{\tau}$ is a conforming vector of parameter estimates for starts of type $\tau$ in Census Region $R$, excluding the intercept from the Census Region equation estimate. Although the Census Region single-family and multifamily starts models are specified as

functions of rates of change in GDP per capita (single-family) or GDP (multifamily), the county level projections were based on calibrated rates of change in disposable income, following the assumption of Wear and Prestemon [20], where the projected rate of change in disposable income was forced to be a constant ratio of the gross output change at the county level. In that way, the Census Region parameter estimates could be applied to the historical county income in calibration and therefore not bias projected starts when modeled on county projections from their study.

The next step was to identify, for each county, the intercept of the resulting county equation that made the average prediction error equal to zero, spanning the 60 quarters from 2001 quarter 1 to 2015 quarter 4 [the estimated quarterly "pseudo permits" for 2000 quarter 1 to 2000 quarter 4 were used in place of lagged values in the quarterly predictions for 2001 using Eqs (5) and (6)].

$$c_{R,m}^{\tau} = \left(\frac{1}{60}\right)\sum_{t=2001,q=1}^{2015,4}\left[\ln\left(h_{t,q,R,m}^{\tau}\right) - \mathbf{x}_{t,q,R,m}^{\tau'}\hat{\boldsymbol{\beta}}_{R}^{\tau}\right] \qquad (6)$$

With an estimate of $c_{R,m}^{\tau}$, for every county in hand, quarterly starts by county could be projected to 2070, given projections of the predictor variables from the Census Region starts models.

Because annual starts are not truly identical to annual permits, the last step was to proportionally adjust every county's predicted single-family and multifamily starts in the projection time frame, 2016 quarter 1 to 2070 quarter 4, to match the Monte Carlo median annual national projected total of single-family and multifamily starts for each quarter, 2016 through 2070.

Data sources for model variables are reported in Table 1.

To project harvested wood products carbon contained in residential housing structures, 2015 to 2070, we used various data, assumptions, and methods based on Smith et al. [34] and McKeever and Howard [35]. The first step in estimating carbon contained in wood being used in housing units was to estimate the quantities of each of five categories of wood products going to single- and multifamily units, including softwood (SW) lumber, hardwood (HW) lumber, SW plywood, oriented strand board (OSB), and non-structural panels that include hardwood plywood, particleboard, medium-density fiberboard, hardboard, and insulation board. Next, the quantities of each of these five categories of wood products used in repair and remodeling were estimated. The estimates of each category of wood products going into construction and into repair and remodeling activities were obtained by estimating wood use intensity (m³/housing unit), based on the historical wood usage by wood products category in single- and multifamily housing units, and the average floor space of housing units constructed

**Table 1. Data sources for housing equation estimates.**

| Variable Name | Data Source |
|---|---|
| Mortgage interest rate | [32] |
| Mortgage delinquency rate | [30, 31] |
| U.S. gross domestic product | [33] |
| U.S. gross domestic product deflator | [33] |
| Housing starts | [28] |
| Housing permits | [28] |
| U.S. population | [17, 29] |
| U.S. population projections by county | [20] |
| U.S. income projections by county | [20] |

**Table 2. Average[1] wood use intensity (m³/housing unit[2]) by wood type, housing type, and usage type used to estimate carbon stored in wood products in use in residential units (source: [35]).**

| Wood product Category | New construction | | Repair & remodeling | |
|---|---|---|---|---|
| | Single-family | Multifamily | Single-family | Multifamily |
| SW lumber | 32.0 [30.92, 32.04, 33.08] | 12.3 [11.64, 12.31, 12.88] | 27.2 [21.06, 27.04, 51.93] | 10.4 [8.2, 10.39, 19.55] |
| HW lumber | 1.2 [1.14, 1.23, 1.29] | 0.4 [0.35, 0.45, 0.5] | 1.1 [0.89, 1.1, 1.34] | 0.4 [0.35, 0.4, 0.41] |
| SW plywood | 6.3 [4.42, 6.27, 8.87] | 2.8 [2.27, 2.84, 3.45] | 6.8 [7.88, 8.6, 14.42] | 3.9 [3.06, 3.9, 7.4] |
| OSB | 20.9 [19.92, 20.88, 21.95] | 5.9 [5.61, 5.9, 6.31] | 12.1 [3.68, 5.5, 10.94] | 1.6 [1.06, 1.56, 3.08] |
| Nonstructural panels[3] | 8.4 [7.18, 8.39, 9.66] | 4.6 [3.84, 4.57, 5.45] | 6.8 [5.25, 6.35, 11] | 3.5 [2.96, 3.46, 5.88] |

[1] The average numbers estimated in this table utilized more recent historical data (from 2000 to 2009) reported in [35] to reflect more recent wood usage trends in residential housing units. The numbers in the square brackets are three parameters [minimum, average, and maximum] of a triangular distribution assumed for the uncertainty analysis.

[2] The average floor area per single-family and multifamily units constructed in the United States, 2000–2009, were 34.96 m² and 17.64 m², respectively [35].

[3] Includes hardwood plywood, particleboard, medium-density fiberboard, hardboard, and insulation board.

each year from 1950 to 2009, as reported in McKeever and Howard [35]. The estimated average wood intensities by wood product category, housing type, and usage type (construction and repair and remodeling) are presented in Table 2.

Carbon stock and stock change (flux) associated with wood used in residential units and landfilled after discard were estimated using the consumption approach [36], which accounts for all wood consumed within U.S. residential housing units, including imported wood products. Consistent with our housing starts projections, the starting year for the estimates of carbon stored in wood products was 2015, where carbon stored in all types of wood products going to all housing units constructed in 2015 was estimated first. The amount of carbon stock remaining in use over time was estimated using the first order decay function (Eq 7, [36]) and assumed half-lives of wood products in single-family and multifamily units (Table 3, [34]):

$$C_{t+1}^i = e^{-k} * C_t^i + \left\lfloor \frac{(1 - e^{-k})}{k} \right\rfloor * inflow(t) \tag{7}$$

Where $t$ = year; $C_t^i$ is the carbon stock in the particular wood product type $i$ (SW lumber, HW lumber, SW plywood, OSB, and nonstructural panels) used in residential units in year $t$

**Table 3. Data and conversion factors used to calculate carbon stored in wood product in use in residential units and in landfills after demolition (source: [34])[1].**

| Variables | Wood product in use | | Wood products in landfills |
|---|---|---|---|
| | Single-family | Multifamily | |
| Half-life of wood products in end uses (yrs) | 100 [95, 100, 105] | 70 [66.5, 70, 73.5] | 14 [13.3, 14, 14.7] |
| Fraction of discarded wood going to landfills | 0.67 [0.64, 0.67, 0.71] | 0.67 [0.64, 0.67, 0.71] | |
| Non degradable fraction of landfilled wood | | | 0.77 [0.66, 0.77, 0.89] |
| Carbon contained in wood products (ton CO₂e/m³) | | | |
| SW lumber | 0.96 [0.92, 0.97, 1.01] | 0.96 [0.92, 0.97, 1.01] | |
| HW lumber | 1.18 [1.13, 1.18, 1.24] | 1.18 [1.13, 1.18, 1.24] | |
| SW plywood | 0.97 [0.92, 0.97, 1.02] | 0.97 [0.92, 0.97, 1.02] | |
| OSB | 1.13 [1.08, 1.13, 1.19] | 1.13 [1.08, 1.13, 1.19] | |
| Nonstructural panels | 1.19 [1.14, 1.19, 1.25] | 1.19 [1.14, 1.19, 1.25] | |

[1] The numbers in the square brackets are three parameters [minimum, average, and maximum] of a triangular distribution assumed for the uncertainty analysis. For half-lives input parameters, the maximum and minimum values are 15% more and less than the average values. For the rest of input parameters, the maximum and minimum values are 5% more and less than the average values.

(beginning in 2015); $k$ is a first-order decay parameter estimated as $k = \ln(2)/HL^i$, where HL is the half-lives of residential units; $inflow(t)$ is the carbon inflow to the particular wood product category in the residential unit in years.

The change in carbon stock between the two periods was estimated as the difference between the next period's carbon stock and the current period's carbon stock:

$$\Delta C_t^i = C_{t+1}^i - C_t^i \tag{8}$$

Carbon stored in wood products remaining in landfills for a given number of years after discard from residential units was estimated following methods suggested by Smith et al. [34]. Briefly, we first estimated the amount of discarded wood at year $t$ after 2015 as the difference in wood remaining in use between two successive years, a fraction (0.67) of which was assumed to end up in landfills. Next, we estimated the quantity of carbon remaining in landfills as a non-degradable pool (77% of discarded carbon), where carbon is permanently sequestered, and as a degradable pool (23% of discarded carbon), where carbon decays based on the first-order decay function (7) with an assumed half-life of 14 years (Table 2).

The assumptions and model parameters used in predicting wood products use in housing and carbon storage are associated with various uncertainties and can have substantial impact on the results (i.e., reliability and credibility of results). Thus, a Monte Carlo simulation approach was used to understand the effects of uncertainties on the estimated average annual wood products carbon stored in residential units, 2015–2070. We assumed a triangular distribution of input parameters (Tables 1 and 2) and the carbon storage model was iterated 5000 times for each SSP.

## Results and discussion

### Equation estimates

Housing starts model estimates are shown in the S10–S20 Tables. To document the significance of population in these new equation estimates, we report models of reduced-form housing starts for both regional total and separate regional single-family and multifamily starts. The separate regional estimates in log-linear form are those used in projecting housing starts, although showing equation estimates for both PPML and log-linear specifications additionally demonstrates the robust effects that population changes have on housing starts in the United States.

S1–S4 Tables report Census Region total starts (single-family and multifamily) equations estimated with PPML methods. Models are all significant as measured by a Wald Chi-squared test, and pseudo-$R^2$'s range from 0.48 (Northeast) to 0.66 (South). These models included the change in the total U.S. population. Notably, the change in U.S. population was positive and statistically significant for total Region starts in the Midwest, South, and West while positive but not significant for the Northeast. The change in real U.S. GDP in these specifications was always significant and positive. Seasonality was evident in all equations as evident by significant quarterly indicator variables, and generally the mortgage delinquency rate and the lagged mortgage interest rate were negatively related to starts as expected, though significance varied by equation.

Log-linear least squares estimates of single-family starts are shown in S5–S8 Tables and PPML estimates are in S13–S16 Tables. The log-linear estimates showed that the total U.S. population was significant at 5% for the Midwest, South, and West regions. Model fits were very high, with $R^2$'s ranging from 0.95 to 0.97, highlighting a highly autoregressive process that included lagged single-family starts also at high significance. And, consistent with the results of Prestemon et al. [11], the log-linear models that included mortgage interest rates had

insignificant serial correlation, as measured by Durbin's H-Statistic. The change in real U.S. GDP per capita in these equations was also highly significant. Unlike the total starts equations for the regions, these equations demonstrated statistically significant and negative relationships between starts and both the mortgage delinquency rate and the lagged mortgage interest rate. Seasonality was common across all, as indexed by quarterly indicator variables.

Multifamily starts models estimated with least squares methods are reported in S9–S12 Tables. Population was not included in these specifications because initial estimates showed insignificance, and so changes in real U.S. GDP were the primary demand driver included in these models. Preliminary versions showed that neither mortgage delinquencies nor mortgage interest rates explained multifamily starts in Census regions, so those two variables were dropped from the specifications (although models that included them fit no better are available from the authors). Whereas models were always statistically significant, the goodness of fit was not as high as it was for single-family starts models, with $R^2$'s ranging from 0.76 to 0.86. Serial correlation was not significant in any model, as measured by Durbin's H-Statistic.

PPML estimates of single-family starts by Census region (S13–S16 Tables) were specified the same as their log-linear counterparts, except that the PPML estimates replaced the total U. S. population with the region's own population. Models fit the data well, all statistically significantly different from a null model and having pseudo-$R^2$'s range from 0.48 to 0.64. Coefficient estimates on mortgage delinquencies and interest rates were negative, where significant. Seasonality was always present, as was an autoregressive process as measured by the coefficient on lagged starts.

Multifamily starts regional PPML estimates are reported in S17–S20 Tables, specified the same as in their log-linear counterparts. We included the mortgage interest rates in these specifications. Here, coefficients on mortgage delinquency rates were either weakly significant and negative or non-significant. Mortgage interest rate coefficient estimates were not significant.

To enable projections to 2070 with our starts models, we needed to estimate time series models of real U.S. GDP, mortgage delinquency rates, and mortgage interest rates that captured both their autoregressivity and their seasonality. Models of the two latter variables were expressed as functions of real U.S. GDP, thereby incorporating the indirect effects of aggregate U.S. income changes on credit access. Prestemon et al. [11] did the same for the first two variables, but because our regional single-family starts models contained the mortgage interest rate, we also developed time series model estimates of this variable. Log-linear least squares model estimates for real GDP are shown in S21 Table and mortgage delinquency rate in S22 Table, both specified similarly to those reported in [11]. The mortgage interest rate model is shown in S23 Table. Clear from S22 and S23 Tables is that mortgage delinquencies and interest rates depend heavily on the rate of U.S. GDP growth.

## Housing starts projections

We evaluated goodness-of-fit out-of-sample of the log-linear versus the PPML model estimates of single-family and multifamily starts and concluded that log-linear versions (S5–S8 Tables for single-family, S9–S12 Tables for multifamily) out-performed the PPML versions (S13–S16 Tables for single-family, S17–S20 Tables for multifamily) in terms of bias and the root mean squared errors of starts. Goodness-of-fit was evaluated with starts models estimated to 2008 quarter 4 and forecast out-of-sample through 2015 quarter 3. We therefore report projections of starts using log-linear specifications of all regional starts models, single-family and multifamily.

Fig 1 shows median (out of 1,000 Monte Carlo iterations) U.S. aggregate single-family starts projections by SSP and also based on historical rates of income and population growth, where

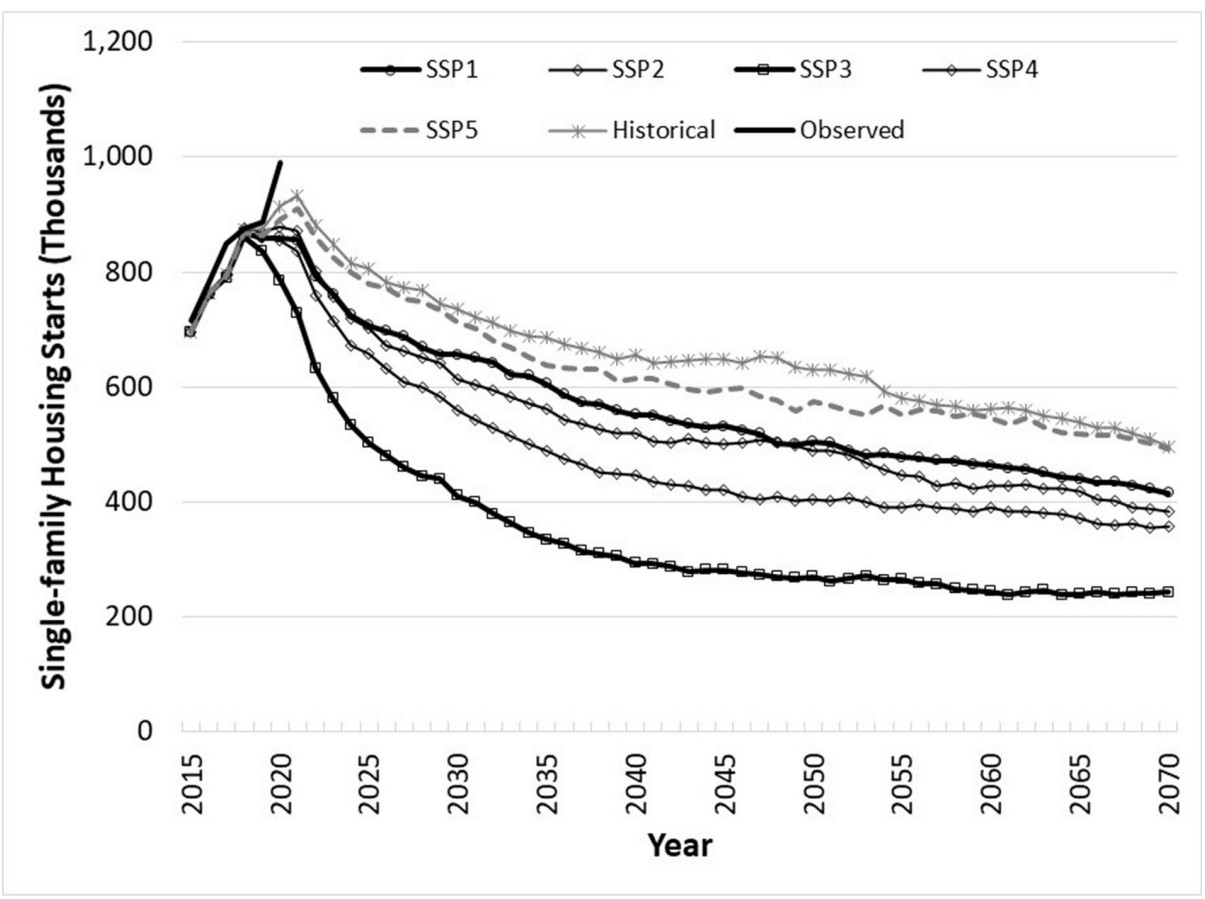

**Fig 1. Annual U.S. single-family housing starts projections, 2015–2070, by Shared Socioeconomic Pathway and historical GDP and population growth rates, based on log-linear regional starts models, summed across median regional levels, based on 1,000 Monte Carlo iterations.**

historical rates of population growth are proxied by the population projection under SSP2. The figure reports the projections, starting in 2016, of the sum of starts across the four Census Regions, the nationwide single-family total. Included in the figure—and in all subsequent figures of starts projections—are observed starts (the solid black line) through 2020. Because historical data on predictors were replaced by projected variables starting in 2019 quarter 1, all starts models make projections from 2019 onward, so that direct comparisons between the Monte Carlo median projected and the observed are possible for 2016–2018 but not warranted for 2019 onward. The lowest single-family starts projected is under SSP3, which has both low economic and population growth (negative population growth after about 2040), with levels settling at less than 250 thousand per year by 2060, barely or even not replacing lost structures. The highest starts are under the "historical" pattern of GDP per capita and population growth and SSP5, and yet those starts even show steadily declining starts to median levels of about 0.5 million/year by 2070.

Multifamily starts nationwide (Fig 2), with projections summed across Census Region projections, indicate that median multi-family starts would range between 200 thousand per year under SSP3 and 300 thousand per year under SSP5. Projection median rates settle at essentially constant levels for each SSP because multifamily starts are driven primarily by economic growth (i.e., GDP).

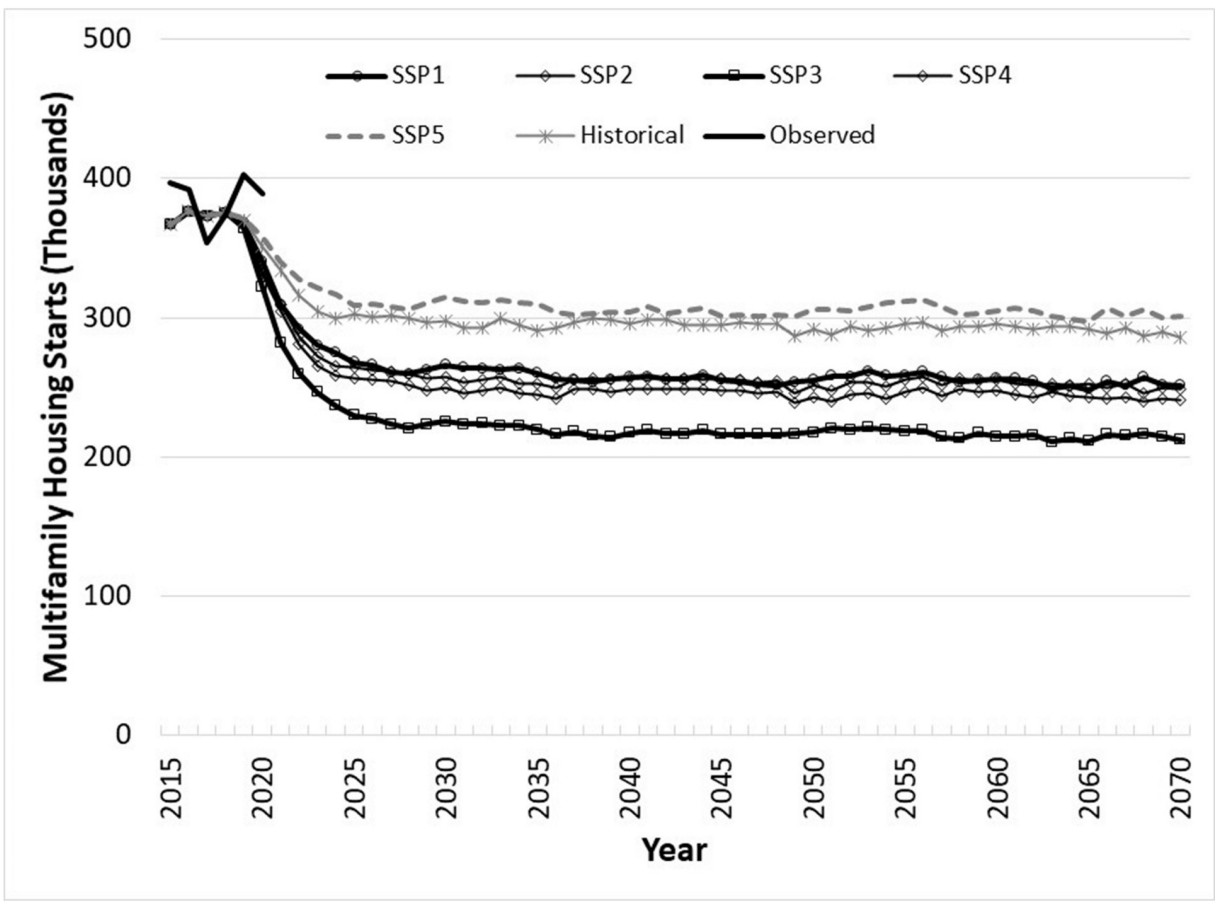

**Fig 2. Annual U.S. multifamily housing starts projections, 2015–2070, by Shared Socioeconomic Pathway and historical GDP and population growth rates, based on log-linear regional starts models, summed across median regional levels, based on 1,000 Monte Carlo iterations.**

Census Region projections of single-family housing starts under the same SSPs and historical population and economic growth rates are shown in the S1 to S8 Figs. In the Northeast (S1 Fig), because population growth rates had already achieved low levels and are projected to remain low, starts levels drop from about 60 thousand/year in the latter half of the 2010s to range from about 20 thousand (SSP3) to 40 thousand (SSP5) by 2070. The Midwest Region (S2 Fig) shows much steeper drops in single-family starts, with median starts levels falling from about 125 thousand/year in the late 2010s to as low as about 15,000 under SSP3 by 2070 but only as high as 50,000/year in 2070 under SSP5. The South Region (S3 Fig) also shows broad declines mirroring the Midwest decline. But because the rate of construction is maintained at 3 to 4 times higher than that of the Midwest, projections show that the South would remain the nation's most active single-family construction market into the foreseeable future. The West (S4 Fig) is intermediate between the Midwest and the South, with median single-family starts dropping from an average of about 200 thousand/year in the late 2010s to between 50 thousand (SSP3) and 120 thousand (SSP5 and historical) by 2070.

S5–S8 Figs show the rapid approach to lower levels of multifamily starts under all SSPs by about 2030 compared to the late-2010s. Rates of multifamily starts are lowest with SSP3 and highest with SSP5 and "historical" scenarios, as was the case with the single-family starts except for the Midwest. The Midwest Region (S6 Fig) has perhaps the narrowest range in multifamily

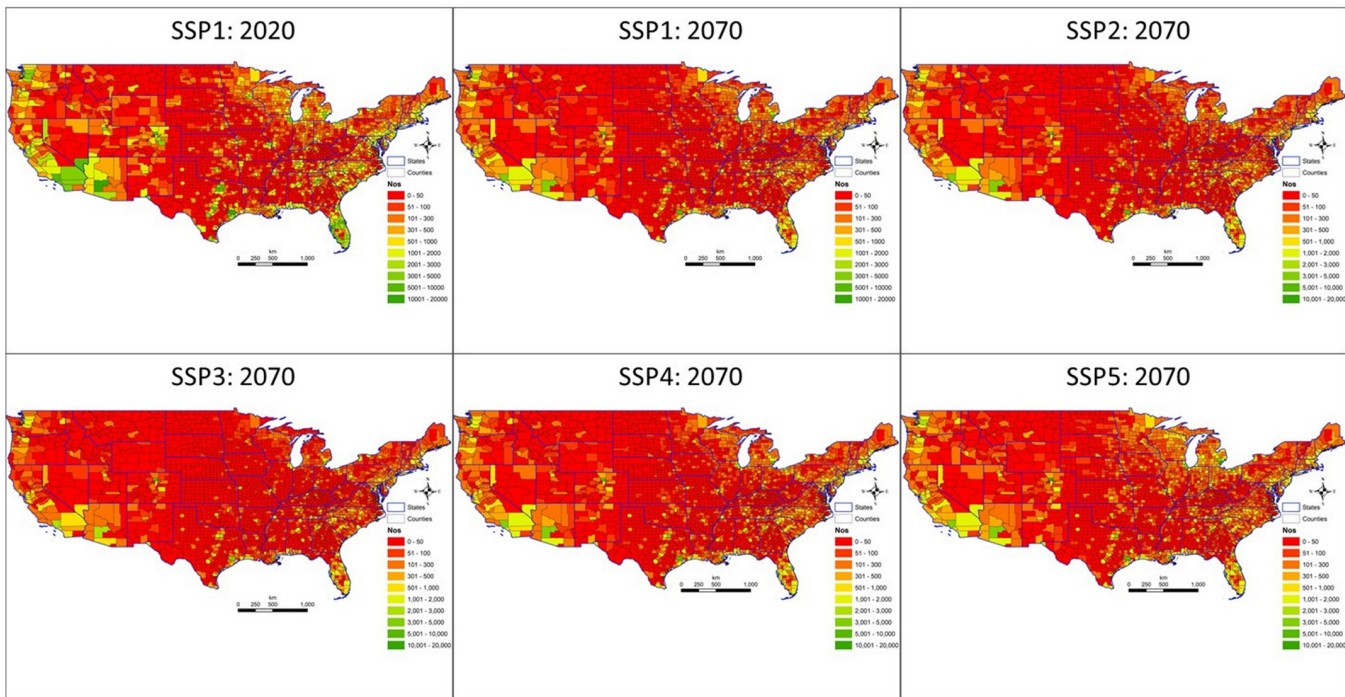

**Fig 3.** Projected housing starts by county in 2020 under Shared Socioeconomic Pathway (SSP) 2 (upper left) and in 2070 for SSP1-5 (upper- mid to lower right). Figure was created by the authors in ESRI's ArcGIS 10.5 software (https://www.esri.com). The USDA-NRCS topo (https://gdg.sc.egov.usda.gov/ GDGHome.aspx) was used as a basemap (NRCS Counties by State, and NRCS States by State) in this figure.

housing starts projected across SSPs, and median projections also indicate that rates in the coming five decades would not differ much from recently observed historical rates. Like single-family starts, multifamily starts are the highest in the South (S7 Fig), followed by the West (S8 Fig), Midwest (S6 Fig), and Northeast (S5 Fig), the same ordering of regional activity shown for single-family starts.

Downscaled projections of housing starts to the county level are shown in Fig 3. The maps report the downscaled county projection of starts that are calibrated to match the national total median starts projected in 2020 under SSP1 (as a reference; projected levels are very similar across all SSPs in 2020, near the beginning of the 2016–2070 projection) and then in 2070 under each of SSP1 through SSP5. One notable feature of the maps is that starts are lower overall in 2070 across all SSPs compared to SSP1 in 2020. Another is that starts growth is projected to be highest in the far southwest (California, Arizona, Nevada), the Gulf Coast, Florida, and the Carolinas across all SSPs, but also including the Lake States under SSP1, 2, and 5. Places where starts are not projected to increase or to decline, under SSP3 especially, in the Great Plains and northern Rockies, are featured in the maps in red. These counties are projected to lose population and to have relatively low economic growth and so do not attract the level of construction expected in faster growth counties.

## Harvested wood products carbon projections

The trajectories of estimated carbon stored in wood products used in residential structures mimic those of the projected housing starts across the nation, the Census Regions and SSPs, with the lowest carbon stocks and changes in stocks projected for SSP3 and in the Northeast region, while the highest of those projected under SSP5 and in the South (Table 4). By 2070,

**Table 4. Total and U.S. Census Region estimates of carbon stocks and average annual changes in carbon stocks (million t CO₂ e) in U.S. single-and multifamily housing units projected to be constructed, 2015–2070, by Shared Socioeconomic Pathway.**

| Census Region | Carbon stock by 2070 | | | | | Average annual changes in carbon stock, 2015–2070 | | | | |
|---|---|---|---|---|---|---|---|---|---|---|
| | SSP1 | SSP2 | SSP3 | SSP4 | SSP5 | SSP1 | SSP2 | SSP3 | SSP4 | SSP5 |
| South | 2,284 | 2,199 | 1,538 | 2,002 | 2,527 | 40 | 39 | 27 | 35 | 45 |
| West | 1,136 | 1,098 | 738 | 991 | 1,233 | 20 | 19 | 13 | 18 | 22 |
| Midwest | 682 | 622 | 344 | 512 | 619 | 12 | 11 | 6 | 9 | 11 |
| Northeast | 414 | 392 | 265 | 353 | 460 | 7 | 7 | 5 | 6 | 8 |
| Total U.S. | 4,517 | 4,311 | 2,885 | 3,858 | 4,839 | 80 | 76 | 51 | 68 | 86 |

combined carbon stocks in wood products remaining in use in single- and multifamily houses projected to be built in the United States, 2015–2070, and those remaining in discarded wood in landfills after the demolition of a structure were shown to range from about 3 billion t CO₂e in SSP3 to 5 billion t CO₂e in SSP5 (Fig 4A). The average annual changes in this stock, 2015–2070, were estimated to range from 51 million t CO₂e in SSP3 to 85 million t CO₂e in SSP5 (Fig 4B and Table 4). To provide a perspective, these figures represent 47% to 78% of total carbon uptake from all wood products in the U.S. in 2019 [4], suggesting that the U.S. residential housing sector would continue to remain the largest HWP carbon sink several decades into the future.

Consistent with the housing starts projections, the South comprises more than 50% of the total U.S. residential housing sector carbon sink in all SSPs, with an estimated average annual change in carbon stock, 2015–2070, of 27 (SSP3) to 45 (SSP5) million t CO₂e, followed by the West (~25%), the Midwest (~15%) and the Northeast (~10%) (Table 4).

Projections at finer scales (county and aggregate state levels) indicate that states such as Texas, Florida, California, Michigan, Ohio, North Carolina, and Georgia would remain among the top 10 contributors to the U.S. residential housing sector carbon sink because of their greater projected housing construction activities, with average annual contributions, 2015–

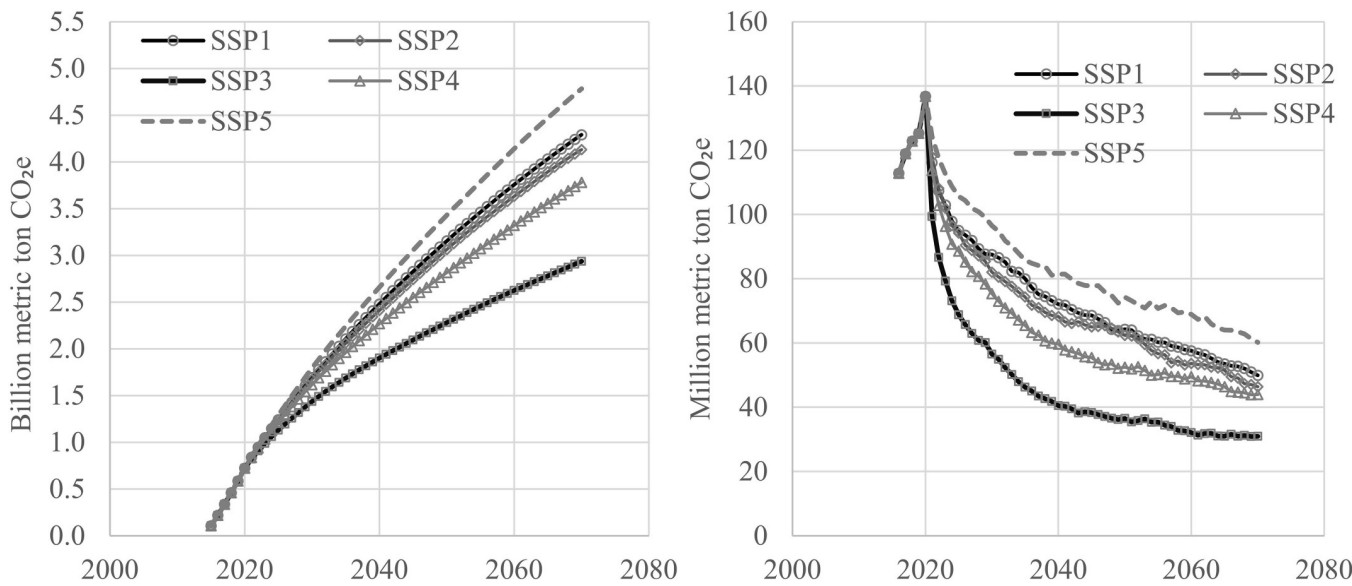

**Fig 4.** Projected carbon stocks (billion t CO₂e) (a) and annual changes in carbon stocks (million t CO₂e) (b) in U.S. single-and multifamily housing units projected to be constructed, 2015–2070, by Shared Socioeconomic Pathway.

**Table 5. Top 25 states with the projected largest contributions to US residential housing sector carbon sink (thousand t CO$_2$e per year), 2015–2070, sorted by Shared Socioeconomic Pathway 5.**

| County and State | Average annual changes in carbon stock, 2015–2070 | | | | |
|---|---|---|---|---|---|
| | SSP1 | SSP2 | SSP3 | SSP4 | SSP5 |
| Texas | 7.31 | 7.17 | 5.58 | 6.78 | 7.25 |
| Florida | 6.29 | 6.15 | 4.70 | 5.79 | 6.22 |
| California | 5.45 | 5.29 | 3.88 | 4.90 | 5.70 |
| Michigan | 3.34 | 3.06 | 1.42 | 2.49 | 4.88 |
| Ohio | 2.48 | 2.28 | 1.12 | 1.88 | 3.56 |
| North Carolina | 3.42 | 3.33 | 2.48 | 3.12 | 3.41 |
| Georgia | 3.13 | 3.06 | 2.27 | 2.86 | 3.12 |
| Colorado | 2.90 | 2.78 | 1.86 | 2.51 | 3.09 |
| Tennessee | 2.50 | 2.43 | 1.76 | 2.26 | 2.51 |
| Wisconsin | 1.77 | 1.64 | 0.84 | 1.36 | 2.49 |
| Washington | 2.31 | 2.23 | 1.58 | 2.05 | 2.46 |
| Illinois | 1.84 | 1.71 | 0.96 | 1.46 | 2.45 |
| New York | 2.22 | 2.17 | 1.72 | 2.07 | 2.15 |
| Indiana | 1.44 | 1.33 | 0.68 | 1.11 | 2.04 |
| South Carolina | 2.00 | 1.95 | 1.43 | 1.82 | 2.01 |
| New Jersey | 1.91 | 1.85 | 1.35 | 1.71 | 1.98 |
| Pennsylvania | 1.92 | 1.88 | 1.46 | 1.79 | 1.81 |
| Arizona | 1.67 | 1.62 | 1.13 | 1.48 | 1.81 |
| Minnesota | 1.30 | 1.21 | 0.65 | 1.02 | 1.80 |
| Louisiana | 1.79 | 1.73 | 1.23 | 1.60 | 1.78 |
| Virginia | 1.75 | 1.71 | 1.27 | 1.60 | 1.73 |
| Missouri | 1.20 | 1.12 | 0.61 | 0.95 | 1.65 |
| Alabama | 1.38 | 1.33 | 0.96 | 1.24 | 1.37 |
| Oregon | 1.23 | 1.18 | 0.82 | 1.08 | 1.33 |
| Maryland | 1.03 | 1.01 | 0.78 | 0.95 | 1.00 |

2070, ranging from 3 to 7 million t CO$_2$e in SSP5 (Table 5 and Fig 5). Looking at the county level projections, counties such as Boulder (Colorado), Harris (Texas), Wayne (Michigan), Maricopa (Arizona), and Los Angeles (California) were shown to be the largest contributors to the U.S. housing sector carbon sink, with more than one million t CO$_2$e stored per year, on average, 2015–2070, in SSP5, in concurrence with greater residential housing construction activities projected for those counties (Fig 5).

Fig 6(A)–6(E) shows the probabilistic distribution (as histogram and cumulative) of annual wood products carbon stored in residential units for each SSP. With 95% certainty, the annual wood products carbon stored in SSP1 is between 74.5 and 87.6 million t CO$_2$e (-7% and +10%). For SSP2, SSP3, SSP4 and SSP5 the annual wood products carbon stored in residential units is 71.9–84.2, 50.1–58.2, 65.6–76.5, and 81.5–95.7 million t CO$_2$e, respectively. Fig 6(F) shows the model input parameters' impacts on the results for SSP2 only. Among all input parameters, the quantities of each of five categories of wood products going to single-family housing units most heavily impact the results. For example, the annual carbon stored in housing units would increase or decrease by 4% if the softwood lumber used in the housing units were to increase or decrease (respectively) by 4%. For all SSP scenarios, the influencing pattern of input parameters on the results is similar, and those details are available from the authors.

Our carbon analyses also considered the potential effects of recycling/reuse of discarded wood products (Table 6). We did additional calculations under Shared Socioeconomic

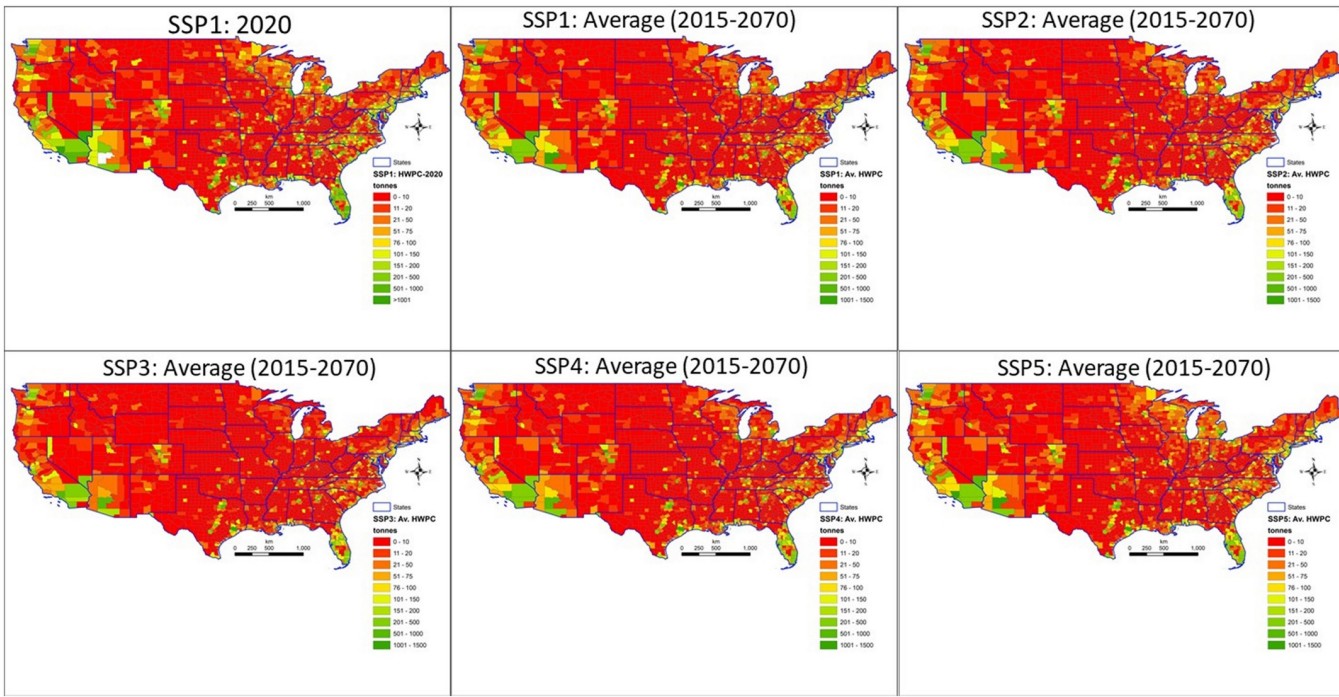

**Fig 5. Projected average annual changes in harvested wood products carbon stocks (thousand t CO$_2$e) contained in U.S. single-and multifamily housing units in use and in landfills after demolition, 2015–2070, by county by Shared Socioeconomic Pathway.** Figure was created by the authors in ESRI's ArcGIS 10.5 software (https://www.esri.com). The USDA-NRCS topo (https://gdg.sc.egov.usda.gov/GDGHome.aspx) was used as a basemap (NRCS Counties by State, and NRCS States by State) in this figure.

Pathway 1 (SSP1), to quantify the effects of recycling on the estimated wood products carbon, using a 17% recycling rate, based on data from US Environmental Protection Agency (EPA) [37] and two rounds of recycling. The additional calculations revealed that recycling discarded wood products for one round would increase the total US residential housing sector wood products carbon stock by 150 million mt CO$_2$e (3.5%) by 2070 and average annual carbon storage (2015–2070) by 2.72 million mt CO$_2$e (3.6%). Evaluated separately for single-family and multifamily units, we found that the percentage contribution to total carbon from recycled wood discarded from multifamily units would be 4.3% higher and single-family units 3.4% higher, although the absolute recycled wood carbon contribution was much larger from single-family units (122 million mt CO$_2$e by 2070) than from multifamily units (28 million mt CO$_2$e by 2070). The relatively higher percentage contribution of recycled wood from multifamily units was due to their assumed lower half-life (70 years) compared to single-family units (100 years), resulting in earlier discard (and recycling) of wood products. Because only a small additional wood quantity would be discarded in the second round of recycling (17% of those discarded in the first round), recycling in the second round generated only a minor additional increase (by 0.1%) in both the carbon stock and average carbon storage over the projection period.

## Conclusions

Our reduced-form estimates of housing starts show that historical starts contrast with projections made with simpler models reported by Prestemon et al. [11], and their differences have implications for projections of stored carbon. In [11], an economic growth rate of 2% would lead to long-run median total starts nationally of slightly less than 1.3 million/year, and 1%

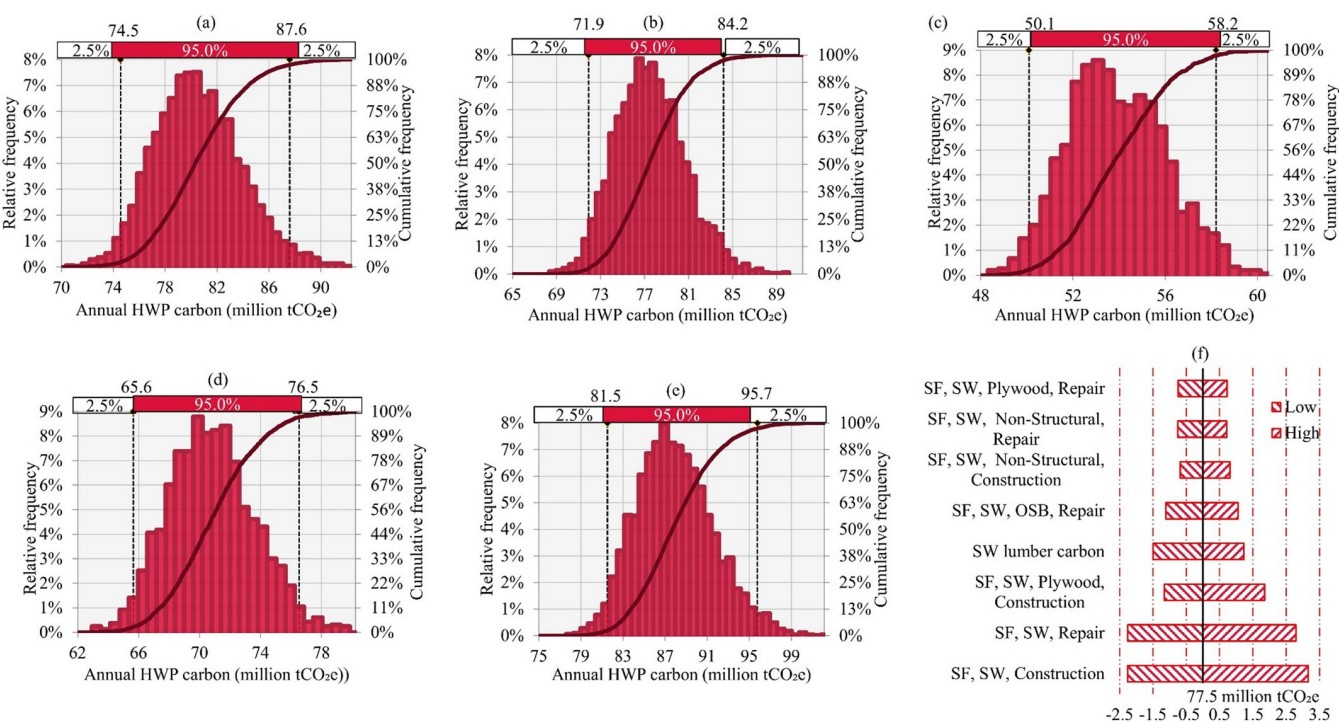

**Fig 6.** Probability distribution (as histogram and cumulative) of annual wood products carbon stored in residential units in use and in landfills, 2015–2070, for SSP 1–5 (a, b, c, d, and e, respectively) and the impact of critical input parameters on the estimated average annual wood products carbon in SSP2 (f).

growth would bring them to about 1 million/year. In this study, we show aggregate total starts (summing starts displayed graphically in Figs 1 and 2) starting out at 1.0 to 1.3 million/year but drifting down over time as projected GDP growth shrinks and population growth declines or turns negative, depending on the SSP in question. The specific addition of population, with its projected slowing to negative growth, depending on the scenario, explains why median national total projected starts are lower by about 0.1 to 0.3 million/year by 2070, compared to [11]. These results are not unexpected, based on evidence from Japan [15, 16], which has in recent years experienced an overall population decline. We contend that the projection models in this research are more accurate than those in [11], based on out-of-sample performance of these models compared to the equivalent models reported in that article. For example, goodness-of-fit out-of-sample (2009q1 to 2015q3) of models that included population show a root mean squared error 24% smaller than models that predicted starts without population. Our models, unlike those in [11], alleviate accuracy distortions of aggregating across single- and multifamily starts: the disaggregation produced reductions in out-of-sample bias (2009q1 to 2015q3) by 93% (and root mean squared by 70%).

Our study has also uncovered the wide degree of spatial heterogeneity in projected trends in construction nationwide, which we have translated into similar patterns of spatial

**Table 6. Total harvested wood products carbon stock by 2070, and average annual carbon storage (2015–2070) in single-family (SF) and multifamily (MF) units estimated with and without recycling of wood products for Shared Socioeconomic Pathway 1 (million metric tons CO₂e).**

| | No recycling | | | Recycling—round 1 | | | Recycling- round 2 | | |
|---|---|---|---|---|---|---|---|---|---|
| | SF | MF | Total | SF | MF | Total | SF | MF | Total |
| Carbon stock by 2070 | 3,632 | 647 | 4,280 | 3,754 | 675 | 4,429 | 3,754 | 676 | 4,432 |
| Average annual carbon storage (2015–2070) | 64.47 | 11.43 | 75.90 | 66.68 | 11.94 | 78.62 | 66.73 | 11.95 | 78.68 |

heterogeneity in growth trends of harvested wood products carbon storage. When the housing starts projections were converted to projections of harvested wood product carbon stored in built structures and landfills, we see that such carbon stocks are projected to increase across much of the United States. The increase suggests that additions to the wood products carbon pool by construction activities more than offsets carbon decay (emissions) resulting from destruction of those structures, which is consistent with starts projections in counties. Despite lower rates of future construction, we find that the U.S. residential housing sector would continue to play an important role in removing carbon from the atmosphere for the next several decades.

Projections of housing starts and harvested wood products carbon generated from our analysis can contribute to refining existing models and in the development of new models of the U.S. forest sector. For example, projections of housing starts generated from our econometric models can serve as inputs to forest sector market models that project changes in U.S. and global demands for forest products (e.g., [38]). Because our models provide county to regional projections of residential housing demand, they can provide the regional demand inputs needed for projections of the inputs demanded by the construction sector at finer than national scales (e.g., [22]). Similarly, projections of multifamily housing starts presented in this study can provide baseline information needed to estimate potential future demand for mass timber, a low-carbon, renewable potential alternative to steel and concrete [39, 40]. In addition, starts projected at the county level offer the opportunity to compare housing growth in wildland-urban interface parts of the United States from this analysis to those using alternative methods (e.g., [41, 42]). Such comparisons may offer new insights on how WUI growth modeling methods might be adjusted to better project development that can impact ecosystems service provisioning [23].

This study offers a framework for development of accurate yet simple models of construction that could be adapted for projections of construction and its impacts at fine spatial scales that may be needed in other countries. Although we focused on the United States and wood-dominated single-family and multifamily housing, prevalent in a limited set of countries with wood-dominated housing such as Canada, the Nordic countries, and Russia, there are carbon consequences of non-wood based construction (which can be potentially more carbon emitting) elsewhere, which could be modeled in similar ways.

Our spatially downscaled projections of starts and harvested wood products carbon in housing reveal where they may be vulnerable to catastrophic risk. Research on climate change and its impacts predict rising rates of wildfire in the United States [24, 43], potentially more damaging hurricanes in the East [25, 44], and rising sea levels along all coasts of the United States (e.g., [26, 45]). A rising number of residential structures may be exposed to these phenomena, potentially accelerating annual rates of housing destruction above the approximate 0.4 million/year observed historically. Although rebuilding following losses would provide new demands for wood, the events themselves would help to contribute to carbon emissions.

The models reported in this study carry with them a set of assumptions that must be acknowledged, each of which could have generated inaccuracies in how housing construction and wood products carbon were projected. First, the projections reported in this study take as given the projections of population and income by county by scenario as reported by Wear and Prestemon [20], whose authors acknowledge the potential limitations of their simple downscaling approach. Importantly, their downscaling models did not directly account for the effects of climate variables or the effects of future changes in climate on demographic shifts, such as those related to rising sea levels impacting coastal counties or rising temperatures that may affect counties differently over time. Likewise, the SSPs presuppose a set of policies influencing population growth—including those affecting fertility rates (e.g., [46–48]), human

longevity (e.g., [49, 50]), and immigration (e.g., [51])—and such policies could have unmodeled heterogeneous impacts across counties of the United States not accounted for in our study.

Our estimates of projected wood products carbon attributed to residential construction activities are based on the average historical sizes (square footage) of single- and multifamily homes and the types and wood usage intensity ($m^3/m^2$) in their construction. To the extent that the average size of homes changes in the future (e.g., due to consumers preferring smaller or larger homes compared to the past) or that future innovation results in new wood products, the quantity and types of wood products going into construction and landfills also would change, rendering our estimates of projected carbon more uncertain. Another factor affecting our carbon results is our assumptions on recycling/reuse practices of discarded wood materials. Although our analysis provided insights into the likely wood products carbon effects of recycling/reuse of wood materials, the analysis did not consider the potential effects on timber harvests from the consequent reduction in the demand for unrecycled wood products, suggesting that the net carbon effects of wood products recycling/reuse activities are more uncertain than reported in this study.

## Supporting information

**S1 Fig. Census Northeast Region single-family housing starts projections, 2015–2070, by Shared Socioeconomic Pathway and historical GDP and population growth rates, based on log-linear regional starts models, summed across median regional levels, based on 1,000 Monte Carlo iterations.**
(TIF)

**S2 Fig. Census Midwest Region Single-family housing starts projections, 2015–2070, by Shared Socioeconomic Pathway and historical GDP and population growth rates, based on log-linear regional starts models, summed across median regional levels, based on 1,000 Monte Carlo iterations.**
(TIF)

**S3 Fig. Census South Region single-family housing starts projections, 2015–2070, by Shared Socioeconomic Pathway and historical GDP and population growth rates, based on log-linear regional starts models, summed across median regional levels, based on 1,000 Monte Carlo iterations.**
(TIF)

**S4 Fig. Census West Region single-family housing starts projections, 2015–2070, by Shared Socioeconomic Pathway and historical GDP and population growth rates, based on log-linear regional starts models, summed across median regional levels, based on 1,000 Monte Carlo iterations.**
(TIF)

**S5 Fig. Census Northeast Region multifamily housing starts projections, 2015–2070, by Shared Socioeconomic Pathway and historical GDP and population growth rates, based on log-linear regional starts models, summed across median regional levels, based on 1,000 Monte Carlo iterations.**
(TIF)

**S6 Fig. Census Midwest Region multifamily housing starts projections, 2015–2070, by Shared Socioeconomic Pathway and historical GDP and population growth rates, based on log-linear regional starts models, summed across median regional levels, based on 1,000**

Monte Carlo iterations.
(TIF)

**S7 Fig. Census South Region multifamily housing starts projections, 2015–2070, by Shared Socioeconomic Pathway and historical GDP and population growth rates, based on log-linear regional starts models, summed across median regional levels, based on 1,000 Monte Carlo iterations.**
(TIF)

**S8 Fig. Census West Region multifamily housing starts projections, 2015–2070, by Shared Socioeconomic Pathway and historical GDP and population growth rates, based on log-linear regional starts models, summed across median regional levels, based on 1,000 Monte Carlo iterations.**
(TIF)

**S1 Table. Northeast U.S. Census Region quarterly total (single-family + multifamily) housing starts, Poisson pseudo-maximum likelihood equation estimates.**
(DOCX)

**S2 Table. Midwest U.S. Census Region quarterly total (single-family + multifamily) housing starts, Poisson pseudo-maximum likelihood equation estimates.**
(DOCX)

**S3 Table. South U.S. Census Region quarterly total (single-family + multifamily) housing starts, Poisson pseudo-maximum likelihood equation estimates.**
(DOCX)

**S4 Table. West U.S. Census Region quarterly total (single-family + multifamily) housing starts, Poisson pseudo-maximum likelihood equation estimates.**
(DOCX)

**S5 Table. Northeast U.S. Census Region quarterly single-family housing starts, least squares equation estimates; dependent variable natural log.**
(DOCX)

**S6 Table. Midwest U.S. Census Region quarterly single-family housing starts, least squares equation estimates; dependent variable natural log.**
(DOCX)

**S7 Table. South U.S. Census Region quarterly single-family housing starts, least squares equation estimates; dependent variable natural log.**
(DOCX)

**S8 Table. West U.S. Census Region quarterly single-family housing starts, least squares equation estimates; dependent variable natural log.**
(DOCX)

**S9 Table. Northeast U.S. Census Region quarterly multifamily housing starts, least squares equation estimates; dependent variable natural log.**
(DOCX)

**S10 Table. Midwest U.S. Census Region quarterly multifamily housing starts, least squares equation estimates; dependent variable natural log.**
(DOCX)

**S11 Table. South U.S. Census Region quarterly multifamily housing starts, least squares equation estimates; dependent variable natural log.**
(DOCX)

**S12 Table. West U.S. Census Region quarterly multifamily housing starts, least squares equation estimates; dependent variable natural log.**
(DOCX)

**S13 Table. Northeast U.S. Census Region quarterly single-family housing starts, Poisson pseudo-maximum likelihood equation estimates.**
(DOCX)

**S14 Table. Midwest U.S. Census Region quarterly single-family housing starts, Poisson pseudo-maximum likelihood equation estimates.**
(DOCX)

**S15 Table. South U.S. Census Region quarterly single-family housing starts, Poisson pseudo-maximum likelihood equation estimates.**
(DOCX)

**S16 Table. West U.S. Census Region quarterly single-family housing starts, Poisson pseudo-maximum likelihood equation estimates.**
(DOCX)

**S17 Table. Northeast U.S. Census Region quarterly multifamily housing starts, Poisson pseudo-maximum likelihood equation estimates.**
(DOCX)

**S18 Table. Midwest U.S. Census Region quarterly multifamily housing starts, Poisson pseudo-maximum likelihood equation estimates.**
(DOCX)

**S19 Table. South U.S. Census Region quarterly multifamily housing starts, Poisson pseudo-maximum likelihood equation estimates.**
(DOCX)

**S20 Table. West U.S. Census Region quarterly multifamily housing starts, Poisson pseudo-maximum likelihood equation estimates.**
(DOCX)

**S21 Table. Least squares regression of the first-difference in the natural logarithm of real U.S. GDP, quarterly, 1984Q1-2014Q3.**
(DOCX)

**S22 Table. Least squares regression of the natural logarithm of the total mortgage delinquency rate in the United States, quarterly, 1984Q1-2014Q3.**
(DOCX)

**S23 Table. Least squares regression of the first-difference in the natural logarithm of the nominal mortgage interest rate, quarterly, 1984Q1-2014Q3.**
(DOCX)

**S1 Data. Projected housing starts by County by Type by Shared Socioeconomic Pathway by Year in the United States.**
(XLSB)

**S2 Data. Projected housing starts by Census Region by Type by Shared Socioeconomic Pathway by Year in the United States.**
(XLSX)

**S3 Data. Projected wood products carbon storage by county by Shared Socioeconomic Pathway by Year in the United States.**
(XLSB)

## Acknowledgments

Authors would like to thank Robert C. Abt and Neelam Poudyal and anonymous reviewers for their review of earlier versions of this manuscript, whose comments helped improve this paper.

**Disclaimer:** The findings and conclusions in this publication are those of the authors and should not be construed to represent any official USDA or U.S. Government determination or policy.

## Author Contributions

**Conceptualization:** Jeffrey P. Prestemon, Prakash Nepal.

**Data curation:** Jeffrey P. Prestemon, Prakash Nepal.

**Formal analysis:** Jeffrey P. Prestemon, Prakash Nepal, Kamalakanta Sahoo.

**Investigation:** Jeffrey P. Prestemon, Prakash Nepal.

**Methodology:** Jeffrey P. Prestemon, Prakash Nepal, Kamalakanta Sahoo.

**Project administration:** Jeffrey P. Prestemon.

**Software:** Jeffrey P. Prestemon, Prakash Nepal, Kamalakanta Sahoo.

**Writing – original draft:** Jeffrey P. Prestemon, Prakash Nepal, Kamalakanta Sahoo.

**Writing – review & editing:** Jeffrey P. Prestemon, Prakash Nepal, Kamalakanta Sahoo.

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
