## [Decision Letter · Decision Letter 0]

25 Mar 2022

PONE-D-22-00404Housing Starts and the Associated Wood Products Carbon Storage by County by Shared Socioeconomic Pathway in the United StatesPLOS ONE

Dear Dr. Prestemon,

Thank you for submitting your manuscript to PLOS ONE. After careful consideration, we feel that it has merit but does not fully meet PLOS ONE’s publication criteria as it currently stands. Therefore, we invite you to submit a revised version of the manuscript that addresses the points raised during the review process.

We look forward to receiving your revised manuscript.

Kind regards,

Andrew T. Carswell

Academic Editor

PLOS ONE

Journal Requirements:

“This research was partially funded by a joint venture agreement between the USDA Forest Service Forest Products Laboratory and the U.S. Endowment for Forestry & Communities, Inc., Endowment Green Building Partnership—Phase 1, no. 18-JV-11111137-021. Authors would like to thank Bob Abt and Neelam Poudyal and anonymous reviewers for their review to the original version of this manuscript, whose comments helped improve this paper.”

“KK was partially funded by a joint venture agreement between the USDA Forest Service Forest Products Laboratory and the U.S. Endowment for Forestry & Communities, Inc., Endowment Green Building Partnership—Phase 1, no. 18-JV-11111137-021. The funders had no role in study design, data collection and analysis, decision to publish, or preparation of the manuscript.”

3. We note that Figure 3 and 5 in your submission contain map images which may be copyrighted. All PLOS content is published under the Creative Commons Attribution License (CC BY 4.0), which means that the manuscript, images, and Supporting Information files will be freely available online, and any third party is permitted to access, download, copy, distribute, and use these materials in any way, even commercially, with proper attribution. For these reasons, we cannot publish previously copyrighted maps or satellite images created using proprietary data, such as Google software (Google Maps, Street View, and Earth). For more information, see our copyright guidelines: http://journals.plos.org/plosone/s/licenses-and-copyright.

 a. You may seek permission from the original copyright holder of Figure 3 and 5 to publish the content specifically under the CC BY 4.0 license. 

Reviewers' comments:

Reviewer's Responses to Questions

**Comments to the Author**

1. Is the manuscript technically sound, and do the data support the conclusions?

Reviewer #1: Yes

Reviewer #2: Yes

2. Has the statistical analysis been performed appropriately and rigorously? 

Reviewer #1: Yes

Reviewer #2: Yes

3. Have the authors made all data underlying the findings in their manuscript fully available?

Reviewer #1: Yes

Reviewer #2: Yes

4. Is the manuscript presented in an intelligible fashion and written in standard English?

Reviewer #1: Yes

Reviewer #2: Yes

5. Review Comments to the Author

Reviewer #1: I was genuinely interested and intrigued with this manuscript, and am mostly in line with accepting it. There are a few things that I wanted to throw out to the authors, however. Here are some follow-up questions:

While I don’t have any disagreement with the statement about population growth being a driver in housing starts, can it not be argued that government policies are huge drivers behind population growth? That seems particularly hard to model, especially out to 2070. Scandinavian countries are heavily incentivizing families to have more children, various countries do the same for migrant labor, and medical breakthroughs continually push out the boundaries of longevity. Since the authors are using already published models providing these projections, this is not a point about which I will quibble with. There is no doubt, however, that the author’s point about market pressures on housing construction is very real, even in the here and now.

The authors do not seem to take into effect the possibility of innovation changes within the construction sector. While the method of constructing homes has been fairly stable for at least the past century or so, there are new and competing technologies that might make traditional wood products in home building somewhat obsolete. OSB, for example, did not begin to be widely used in the construction trade until the 1970s. Similar types of innovations can come along to disrupt the mix of wood products appropriately aid out in lines 223-225.

In the demand model laid out at the beginning of the methods section, I am assuming that “credit” is not just an estimate of the issuance of credit through robustness of one’s credit scores, but is also a function of wealth characteristics over time to cover the down payment (in homeownership situations).

On lines 201-202, you mentioned a difference in the income variables within both the single-family and multi-family models. Why does one utilize disposable income and the other income?

Does the model also take into effect changes over time in consumer preferences for house SIZE, another determinant ultimately in board feet used per construction project? If not, please explain the reasoning there. One could argue that this trend could go either way in the next 50 years. Historically speaking, the average square footage per housing unit has gone up over time as demand for housing increases with household purchasing power and shifts in consumer preference toward housing. At the same time, however, generational differences between boomers and say, Gen Zers, shows that the younger crowd seems to buy into the concept of more sustainable living patterns (including reduced housing footprint).

Also, does the model incorporate the carbon storage for recycled lumber products, rather than having the lumber stock go to waste? I have heard of firms utilizing this strategy as part of an overall sustainability strategy, but do not know how prevalent it is both now and into the future. I also do not know if it makes a lot of difference either.

While climate change was a primary driver of the study (and appropriately so), it was not very clear to me that the authors used climate change as an impetus for the demographic shifts in this country. Is that an appropriate statement? Couldn’t climate change create a situation in which Minnesota and the Dakotas become the ideal places to live and grow food by 2070? (Just an example)

I did not see any mention by the authors regarding limitations of their study. It seems likely that there would be some.

Finally, do the models included within this research have generalizability to other countries besides the U.S.? I noticed during discussion of the lit review that other countries’ growth models (Japan’s, for one) were used toward the development and design of this research…what about the reverse?

Reviewer #2: The author(s) have presented a detailed study that is conducted in the United States on housing and the associated wood product carbon. The manuscript is well written and structured. However, there are some areas that require improvement:

First, the abstract of the manuscript should be rewritten. This is the first part of the manuscript, and it is important to clearly inform potential readers the main problem, the aim of the study, the methodology, the practical and theoretical implications of the study. Before these, it is important to briefly state the general scope of the study before narrowing down to the specifics of the study. Similarly, the research problem should be clearly expressed in the introduction.

Keywords have not been included in the manuscript.

Next is the conclusion. The conclusion is too lengthy. It reads like another discussion. The author(s) could consider restructuring the conclusion for brevity and clarity.

6. PLOS authors have the option to publish the peer review history of their article (what does this mean?). If published, this will include your full peer review and any attached files.

Reviewer #1: No

Reviewer #2: No

---

## [Author Response · Author response to Decision Letter 0]

26 Apr 2022

Response to the Reviewers’ comments

Manuscript # PONE-D-22-00404

Manuscript Title: Housing Starts and the Associated Wood Products Carbon Storage by County by Shared Socioeconomic Pathway in the United States

Editor’s Comments

Response: We have revised the formatting according to the referenced requirements. 

and

Response: We have revised the author listing and affiliations accordingly.

“This research was partially funded by a joint venture agreement between the USDA Forest Service Forest Products Laboratory and the U.S. Endowment for Forestry & Communities, Inc., Endowment Green Building Partnership—Phase 1, no. 18-JV-11111137-021. Authors would like to thank Bob Abt and Neelam Poudyal and anonymous reviewers for their review to the original version of this manuscript, whose comments helped improve this paper.”

Please remove any funding-related text from the manuscript and let us know how you would like to update your Funding Statement. 

Response: We have deleted the funding information from the revised Acknowledgments section, as required.

Currently, your Funding Statement reads as follows:

“KK was partially funded by a joint venture agreement between the USDA Forest Service Forest Products Laboratory and the U.S. Endowment for Forestry & Communities, Inc., Endowment Green Building Partnership—Phase 1, no. 18-JV-11111137-021. The funders had no role in study design, data collection and analysis, decision to publish, or preparation of the manuscript.”

Response: We have included the funding statement within the revised cover letter. Thank you for uploading the funding statement online. We need to also request that, rather than “KK,” our third author be referred to as “Dr. Kamalakanta Sahoo.”

3. We note that Figure 3 and 5 in your submission contain map images which may be copyrighted. All PLOS content is published under the Creative Commons Attribution License (CC BY 4.0), which means that the manuscript, images, and Supporting Information files will be freely available online, and any third party is permitted to access, download, copy, distribute, and use these materials in any way, even commercially, with proper attribution. For these reasons, we cannot publish previously copyrighted maps or satellite images created using proprietary data, such as Google software (Google Maps, Street View, and Earth). For more information, see our copyright guidelines: http://journals.plos.org/plosone/s/licenses-and-copyright.

 a. You may seek permission from the original copyright holder of Figure 3 and 5 to publish the content specifically under the CC BY 4.0 license. 

Response: The figures are original and created in ArcGIS software 10.5. The Map was taken from the USDA-NRCS Geospatial data gateway (https://gdg.sc.egov.usda.gov/GDGOrder.aspx). All other data used to generate the figures are from this research. We examined other articles published in PLOS ONE, and following them, have updated the figure captions for figures 3 and 5 to include reference the ESRI ArcGIS 10.5 software and the USDA-NRCS topo basemap. 

Response: We did not remove (retract) any previously cited references. We added the following: Mayer et al. (2009), Ebenstein et al. (2017), Dwyer-Lindgren et al. (2017), Massey and Pren (2012), de Silva and Tenreyro (2017), Kearney and Levine (2015), Kearney et al. (2022), USEPA (2020), so that we could fully respond to comments of reviewers 1 and 2.  

Reviewer #1: I was genuinely interested and intrigued with this manuscript, and am mostly in line with accepting it. There are a few things that I wanted to throw out to the authors, however. Here are some follow-up questions:

Response: Thank you for your constructive comments. We took your comments seriously and addressed all of them to the fullest extent possible. 

1. While I don’t have any disagreement with the statement about population growth being a driver in housing starts, can it not be argued that government policies are huge drivers behind population growth? That seems particularly hard to model, especially out to 2070. Scandinavian countries are heavily incentivizing families to have more children, various countries do the same for migrant labor, and medical breakthroughs continually push out the boundaries of longevity. Since the authors are using already published models providing these projections, this is not a point about which I will quibble with. There is no doubt, however, that the author’s point about market pressures on housing construction is very real, even in the here and now.

Response: We agree that we take as given the population (and income) growth projections provided by the SSP scenarios, which themselves presuppose a set of policies affecting fertility rates, longevity, and immigration. We also must acknowledge the potential limitations of the data downscaling models of Wear and Prestemon, which ignore potential spatial heterogeneity in the effects of government policies on fertility, longevity, and immigration. To directly address your comment, in the revised text, we acknowledge and recognize these points with additional citations of relevant research on these issues (lines 573-579). 

Additional literature cited:

Ebenstein A, Fan M, Greenstone M, He G, Zhou M. New evidence on the impact of sustained exposure to air pollution on life expectancy from China’s Huai River Policy. Proc Natl Acad Sci U S A. 2017;114(39):10384-9. doi: 10.1073/pnas.1616784114

Dwyer-Lindgren L, Bertozzi-Villa A, Stubbs RW, Morozoff C, Mackenbach JP, van Lenthe FJ, et al. Inequalities in Life Expectancy Among US Counties, 1980 to 2014: Temporal Trends and Key Drivers. JAMA Intern Med. 2017;177(7):1003-1011. doi: 10.1001/jamainternmed.2017.0918

Massey DS, Pren KA. Unintended consequences of US immigration policy: explaining the post-1965 surge from Latin America. Popul Dev Rev. 2012;38(1):1-29. doi: 10.1111/j.1728-4457.2012.00470.x 

de Silva T, Tenreyro S. Population control policies and fertility convergence. J Econ Perspec. 2017;31(4):205-228. doi: 10.1257/jep.31.4.205

Kearney MS, Levine PB. Investigating recent trends in the U.S. teen birth rate. J Health Econ. 2015;41:15-29. doi: 10.1016/j.jhealeco.2015.01.003

Kearney MS, Levine PB, Pardue L. The puzzle of falling US birth rates since the Great Recession. J Econ Perspec. 2022;36(1):151-176. doi: 10.1257/jep.36.1.151

2. The authors do not seem to take into effect the possibility of innovation changes within the construction sector. While the method of constructing homes has been fairly stable for at least the past century or so, there are new and competing technologies that might make traditional wood products in home building somewhat obsolete. OSB, for example, did not begin to be widely used in the construction trade until the 1970s. Similar types of innovations can come along to disrupt the mix of wood products appropriately laid out in lines 223-225.

Response: The key focus of our analysis was developing parsimonious but robust statistical models to project numbers of single-family and multifamily units likely to be constructed in the United States under varying socioeconomic growth scenarios, which are not necessarily affected by products innovations. However, we agree that the quantity and mixes of wood products going to be used in those units might be affected by future innovation in wood products, and so are the estimated wood products carbon contained in those housing units. The uncertainty analyses that we carried out considers to some extent such possibility of future changes in wood product mixes. For example, our uncertainty analyses consider potential changes in the half-life of wood products in end uses (which can increase if more durable wood products are invented in future) and discard rate of wood in landfills (which can decrease if more durable wood products are invented in future. However, we do not have enough information to be able to quantify likely innovation and likely new wood products mixes going to be used in housing units in future. We highlighted these facts in the Conclusions section of our revised manuscript as follows (lines 580-585):

“Our estimates of projected wood products carbon attributed to residential construction activities are based on the average historical sizes (square footage) of single- and multifamily homes and the types and wood usage intensity (m3/m2) in their construction. To the extent that the average size of homes changes in the future (e.g., due to consumers preferring smaller or larger homes compared to the past) or that future innovation results in new wood products, the quantity and types of wood products going into construction and landfills also would change, rendering our estimates of projected carbon more uncertain.”

3. In the demand model laid out at the beginning of the methods section, I am assuming that “credit” is not just an estimate of the issuance of credit through robustness of one’s credit scores, but is also a function of wealth characteristics over time to cover the down payment (in homeownership situations).

Response: In the model specified in this study, credit factors in equations (1) and (3) are assumed to be unbiasedly captured by the mortgage delinquency rate and the mortgage interest rate, and separately modeled equations of those two variables are used to project housing starts. The delinquency rate is modeled as a function of GDP and thereby indirectly captures wealth; that equation explained 97% of variation in the historical series of the mortgage delinquency rate (Supplemental table S22). The mortgage interest rate equation also is modeled as a function of GDP and so also captures wealth. The delinquency rate, following Prestemon et al. (2018) and consistent with Mayer et al. (2009), is also intended to, in combination with the mortgage interest rate, summarize credit access. To directly address your comment, we emphasize in the revised text (lines 144-147, 347-354) how those two variables are intended to capture the effects of wealth characteristics and mortgage credit accessibility.

Additional literature cited:

Mayer C, Pence K, Sherlund SM. The rise in mortgage defaults. J Econ Perspect. 2009;23(1):27-50. doi: 10.1257/jep.23.1.27

4. On lines 201-202, you mentioned a difference in the income variables within both the single-family and multi-family models. Why does one utilize disposable income and the other income?

Response: In the statistical models, we modeled Census Region single-family starts as a function of the rate of change in GDP per capita and multifamily starts as a function of the rate of change in GDP. Although the Census Region single-family and multifamily starts models are specified as functions of rates of change in GDP per capita (single-family) or GDP (multifamily), in the county level projections, although calibrated on rates of change in disposable income, following the assumption of Wear and Prestemon (2019), the projected rate of change in disposable income was forced to be a constant ratio of the gross output change at the county level. In that way, the Census Region parameter estimates could be applied to the historical county income in calibration and therefore not bias projected starts when modeled on county from their study. To directly address this omission, we now explain this assumption around the expression of equation (5) (lines 211-217).

5. Does the model also take into effect changes over time in consumer preferences for house SIZE, another determinant ultimately in board feet used per construction project? If not, please explain the reasoning there. One could argue that this trend could go either way in the next 50 years. Historically speaking, the average square footage per housing unit has gone up over time as demand for housing increases with household purchasing power and shifts in consumer preference toward housing. At the same time, however, generational differences between boomers and say, Gen Zers, shows that the younger crowd seems to buy into the concept of more sustainable living patterns (including reduced housing footprint).

Response: We added the following text to address this potential shortcoming of our modeling approach (lines 580-585): 

“Our estimates of projected wood products carbon attributed to residential construction activities are based on the average historical sizes (square footage) of single- and multifamily homes and the types and wood usage intensity (m3/m2) in their construction. To the extent that the average size of homes changes in the future (e.g., due to consumers preferring smaller or larger homes compared to the past) or that future innovation results in new wood products, the quantity and types of wood products going into construction and landfills also would change, rendering our estimates of projected carbon more uncertain.”

6. Also, does the model incorporate the carbon storage for recycled lumber products, rather than having the lumber stock go to waste? I have heard of firms utilizing this strategy as part of an overall sustainability strategy, but do not know how prevalent it is both now and into the future. I also do not know if it makes a lot of difference either.

Response: Thank you for bringing the issue of recycling wood products. The information on the amounts of wood products going to be recycled and the number of rounds they are recycled are scarce and/or less certain. Nevertheless, we did additional calculations under the shared socioeconomic pathway 1 (SSP1) to get a sense of difference on the estimated wood products carbon with and without recycle, using a 17% recycle rate based on the US Environmental Protection Agency (EPA) data (US EPA 2020) and two rounds of recycle. The additional calculations revealed that recycling all products for one round would increase the total US residential housing sector wood products carbon stock by 150 million mt CO2e (3.5%) by 2070, and average annual carbon storage (2015-2070) by 2.72 million mt CO2e (3.6%). Evaluated separately for single-family and multifamily units, we found that the percentage contribution to total carbon from recycled wood discarded from multifamily units would be higher (4.3%) compared to single-family units (3.4%), although the absolute recycled wood carbon contribution was much higher from single-family units (122 million mt CO2e by 2070) compared to multifamily units (28 million mt CO2e by 20). The relatively higher percentage contribution of recycled wood from multifamily unit was due to assumed lower half-life (70 years vs 100 years for single family) resulting into earlier discard (and recycle) of wood products. Recycling wood products for the second round resulted into slight additional increase (0.1%) in both carbon stock and average carbon storage during our projection period. This is because very little amount of wood would be discarded in the second round (17% of those discarded in the first round). We have provided a table summarizing these results in a new table (Table 6).

Table 6. Total harvested wood products carbon stock by 2070, and average annual carbon storage (2015-2070) in single-family (SF) and multifamily(MF) units with and without considering recycling of wood products for Shared Socioeconomic Pathway 1 (million metric tons CO2e). 

 No Recycling Recycling - round 1 Recycling- round 2

 SF MF Total SF MF Total SF MF Total

Carbon stock by 2070 3,632 647 4,280 3,754 675 4429 3,754 676 4432

Average annual carbon storage (2015-2070) 64.47 11.43 75.90 66.68 11.94 78.62 66.73 11.95 78.68

7. While climate change was a primary driver of the study (and appropriately so), it was not very clear to me that the authors used climate change as an impetus for the demographic shifts in this country. Is that an appropriate statement? Couldn’t climate change create a situation in which Minnesota and the Dakotas become the ideal places to live and grow food by 2070? (Just an example)

Response: The projections reported in this study take as given the projections of population and income by county by scenario as reported by Wear and Prestemon (2019). Those authors acknowledge the potential limitations of their simple downscaling approach, which does not directly account for the effects of climate variables and the effects of future changes in climate on demographic shifts. The authors of that article indicate that their model (Wear and Prestemon 2019, p. 15):

“…is derived from a simple concept, labor-driven migration, resulting in gradual reductions in per capita income variation across space, augmented by agglomeration dynamics. Notably, this model does not account for the influence of climate change projections on potential long run change in population or economic growth. While historical climate influences are embedded in the model and might be evident in near term projections, longer term movements in populations in the United States may be structurally influenced by climate change—e.g., by sea level rise in coastal counties of the East and temperature increases in the Southwest…” 

To directly acknowledge this possible weakness in the projection data used in this study, we have inserted additional text along these lines in the concluding portions of the manuscript text (lines 571-579). 

8. I did not see any mention by the authors regarding limitations of their study. It seems likely that there would be some.

Response: We have highlighted the following additional limitations and uncertainties in the revised manuscript (see lines 569-590):

1. Described the limitations of the county projections’ direct modeling of the effects of climate change on demographic shifts and hence construction (see response to comment 7). 

2. Clarified how policies affect population growth rates through impacts on fertility, mortality, and immigration, and how the SSPs, which embody those policies through their population projections, were not similarly downscaled by Wear and Prestemon in a way that might have affected the spatial distribution of their impacts within the United States. 

3. Highlighted the potential effects on estimated carbon of future innovation in wood products (see response above).

4. Highlighted the effects on estimated carbon of future changes in square footage and wood usage in housing units (see response above).

9. Finally, do the models included within this research have generalizability to other countries besides the U.S.? I noticed during discussion of the lit review that other countries’ growth models (Japan’s, for one) were used toward the development and design of this research…what about the reverse?

Response: With regard to the models of housing starts: The results are relevant to other countries who have information on fine-scale projected changes in population and income and need to project/predict where new construction is likely to be concentrated and where it is likely to diminish as populations change. With information on historical construction at fine scales and estimate of aggregate relationships between income and population, the same fine-scale projections of construction could be done. Although the focus in the study was the United States and wood-dominated single-family and multifamily housing, which is also prevalent in Canada, Nordic countries, and Russia, there are carbon consequences of non-wood based construction (which can be potentially more carbon emitting) elsewhere, which could be modeled in similar ways. And with respect to the HWP C model: Methods related to estimating carbon in harvested wood products in residential units can be used for other countries, but they need to be adapted to actual country-level data on the specific variables (e.g., half-life, landfill rate, average square footage, wood type and amounts per floor space). In the revised paper, we make these points in the concluding paragraphs of the revised manuscript (lines 555-560).

Reviewer #2: The author(s) have presented a detailed study that is conducted in the United States on housing and the associated wood product carbon. The manuscript is well written and structured. However, there are some areas that require improvement:

First, the abstract of the manuscript should be rewritten. This is the first part of the manuscript, and it is important to clearly inform potential readers the main problem, the aim of the study, the methodology, the practical and theoretical implications of the study. Before these, it is important to briefly state the general scope of the study before narrowing down to the specifics of the study. Similarly, the research problem should be clearly expressed in the introduction.

Response: We have revised the abstract as suggested. 

Keywords have not been included in the manuscript.

Response: Although keywords are required in the online submission, which we did provide in that process, they are not part of contents of the manuscript text required by the journal. We ensured in our revised submission process that our keywords were indeed listed. 

Next is the conclusion. The conclusion is too lengthy. It reads like another discussion. The author(s) could consider restructuring the conclusion for brevity and clarity.

Response: We have endeavored to revise the conclusions to be more concise. To respond to another reviewer, however, we had to add new discussion of caveats. We hope that you are satisfied with the changes made.

---

## [Decision Letter · Decision Letter 1]

2 Jun 2022

Housing Starts and the Associated Wood Products Carbon Storage by County by Shared Socioeconomic Pathway in the United States

PONE-D-22-00404R1

Dear Dr. Prestemon,

We’re pleased to inform you that your manuscript has been judged scientifically suitable for publication and will be formally accepted for publication once it meets all outstanding technical requirements.

Kind regards,

Andrew T. Carswell

Academic Editor

PLOS ONE

Additional Editor Comments (optional):

Reviewers' comments:

Reviewer's Responses to Questions

**Comments to the Author**

1. If the authors have adequately addressed your comments raised in a previous round of review and you feel that this manuscript is now acceptable for publication, you may indicate that here to bypass the “Comments to the Author” section, enter your conflict of interest statement in the “Confidential to Editor” section, and submit your "Accept" recommendation.

Reviewer #1: All comments have been addressed

Reviewer #2: All comments have been addressed

2. Is the manuscript technically sound, and do the data support the conclusions?

Reviewer #1: Yes

Reviewer #2: Yes

3. Has the statistical analysis been performed appropriately and rigorously? 

Reviewer #1: Yes

Reviewer #2: Yes

4. Have the authors made all data underlying the findings in their manuscript fully available?

Reviewer #1: Yes

Reviewer #2: Yes

5. Is the manuscript presented in an intelligible fashion and written in standard English?

Reviewer #1: Yes

Reviewer #2: Yes

6. Review Comments to the Author

Reviewer #1: I appreciate all of the work that the author team has put in to ensure the quality of this manuscript.

Reviewer #2: (No Response)

7. PLOS authors have the option to publish the peer review history of their article (what does this mean?). If published, this will include your full peer review and any attached files.

Reviewer #1: No

Reviewer #2: No

---

## [Editor Report · Acceptance letter]

26 Jul 2022

PONE-D-22-00404R1 

Housing Starts and the Associated Wood Products Carbon Storage by County by Shared Socioeconomic Pathway in the United States 

Dear Dr. Prestemon:

I'm pleased to inform you that your manuscript has been deemed suitable for publication in PLOS ONE. Congratulations! Your manuscript is now with our production department. 

Kind regards, 

on behalf of

Dr. Andrew T. Carswell 

Academic Editor

PLOS ONE